# Efficient Asynchronous Federated Learning for AUV Swarm

**DOI:** 10.3390/s22228727

**Published:** 2022-11-11

**Authors:** Zezhao Meng, Zhi Li, Xiangwang Hou, Jun Du, Jianrui Chen, Wei Wei

**Affiliations:** 1School of Mechano-Electronic Engineering, Xidian University, Xi’an 710071, China; 2Department of Electronic Engineering, Tsinghua University, Beijing 100084, China; 3Tsinghua Shenzhen International Graduate School, Tsinghua University, Shenzhen 514231, China

**Keywords:** federated learning (FL), autonomous underwater vehicle (AUV), gradient compression, communication resource optimization, proximal policy optimization 2 (PPO2)

## Abstract

The development of automatic underwater vehicles (AUVs) has brought about unprecedented profits and opportunities. In order to discover the hidden valuable data detected by an AUV swarm, it is necessary to aggregate the data detected by AUV swarm to generate a powerful machine learning model. Traditional centralized machine learning generates a large number of data exchanges and faces problems of enormous training data, large-scale models, and communication. In underwater environments, radio waves are strongly absorbed, and acoustic communication is the only feasible technology. Unlike electromagnetic wave communication on land, the bandwidth of underwater acoustic communication is extremely limited, with the transmission rate being only $1/10^{5}$ of the electromagnetic wave. Therefore, traditional centralized machine learning cannot support underwater AUV swarm training. In recent years, federated learning could only interact with model parameters without interacting with data, which greatly reduced communication costs. Therefore, this paper introduces federated learning into the collaboration of an AUV swarm. In order to further reduce the constraints of underwater scarce communication resources on federated learning and alleviate the straggler effect, in this work, we designed an asynchronous federated learning method. Finally, we constructed the optimization problem of minimizing the weighted sum of delay and energy consumption, relying on jointly optimizing the AUV CPU frequency and signal transmission power. In order to solve this complex optimization problem of high-dimensional non-convex time series accumulation, we transformed the problem into a Markov decision process (MDP) and use the proximal policy optimization 2 (PPO2) algorithm to solve this problem. The simulation results demonstrate the effectiveness and superiority of our method.

## 1. Introduction

An automatic underwater vehicle (AUV) is a kind of submarine robot that can carry out ocean sampling activities independently; it is widely used in underwater research and the marine industry. For example, AUVs have been used to sample coastal frontiers, monitor coastal areas, measure thermocline turbulence, obtain interdisciplinary data, conduct fishery research, and install submarine cables under the frozen sea [1,2]. However, due to the limited capacity of a single AUV, processing more complex tasks by AUV swarm has become a hot research topic. In recent years, underwater research based on the AUV group has become more popular [3,4]. The advantages of the AUV swarm cooperation over a single AUV can be summarized into two points. On the one hand, an AUV swarm can obtain more data than a single AUV, such as underwater hydrological characteristic modeling, on the other hand, an AUV swarm can perform tasks that cannot be completed by a single AUV, such as rounding-up targets, formation cruising, collaborative positioning, etc. In the above two aspects, it is often critical to establish a powerful machine learning model to mine the values behind the observed data, or to improve the effectiveness of the AUV’s own actions. Specifically, for the sake of exploring the hidden valuable information of the data detected by the AUV swarm, it is necessary to aggregate the data detected by the AUV swarm to generate a powerful machine learning model, such as forming the hydrological characteristics model of the water area by data fitting. Moreover, in order to improve the movement of the AUV, it is a requisite to train the AUV to obtain a strong reinforcement learning model to achieve efficient autonomous action of the AUV. In order to support the above two goals, traditional centralized machine learning needs to interact with a large number of data, which will require a lot of communication resources. However, unlike on land, underwater electromagnetic wave communications with high bandwidths cannot be used; they often rely on underwater acoustic communication for data transmission. The bandwidth of underwater acoustic communication is very limited, and the transmission rate is only $1/10^{5}$ of the electromagnetic wave. Therefore, traditional centralized machine learning cannot support underwater AUV swarm training and the production of large models. Federated learning, a new machine learning framework, was first proposed by Google in 2016 [5]. Each data holder uses his/her own data to train a model, and the models interact with each other. Finally, a global model is obtained through model aggregation. Unlike traditional centralized machine learning, federated learning can only interact model parameters without interacting with data, which can greatly reduce communication overhead and reduce dependence on communication resources, and is suitable for the environment where underwater communication resources are scarce. Therefore, this paper introduces federated learning into the collaboration of the AUV swarm. Furthermore, considering that the interaction model parameters are much less than the interactions with the original data, there is still a large number of data, and the traditional federated learning method (i.e., FEDAVG proposed by Google [5]) is synchronized, which is prone to the emergence of the straggler effect, especially in unreliable underwater information transmission and large time delays. Therefore, this paper designs an asynchronous federated learning method to further reduce data transmission and reduce the straggler effect, which will play an important role in the harsh underwater environment. Furthermore, considering the limited energy of the AUV and the time requirement of model training, we constructed an optimization problem of minimizing the weighted sum of delay and energy consumption, relying on jointly optimizing AUV CPU frequency and signal transmission power. In order to solve this complex optimization problem of high-dimensional non-convex time series accumulation, we built the problem into an MDP and used the PPO2 algorithm to solve this problem. The main contributions are as follows:To the authors’ knowledge, this paper is the first to introduce federated learning into the AUV swarm. Federated learning can help AUV swarm train large-scale machine learning models in an environment where underwater communication resources are scarce.In order to further reduce the constraints of underwater scarce communication resources on federated learning, we designed an asynchronous federated learning method, which can effectively alleviate the straggler effect and further reduce data interaction.We constructed the optimization problem of minimum delay and energy consumption, by jointly optimizing AUV CPU frequency and signal transmission power.In order to solve the optimization problem efficiently, we converted it into an MDP and proposed the PPO2 algorithm to solve this problem. The simulation results verify the effectiveness of the proposed algorithm.

The content and organizational structure of this article are as follows:In the third section, a federated learning model based on the AUV swarm is established. First, we established a federal learning model. Second, the communication system model was established. Then the control model was established, the nodes participating in this round of upload were selected locally, and redundant nodes were adaptively skipped. Then a delay model was established to calculate the time and total time of each phase of federated learning. Then the energy consumption model was established as the energy constraint. Finally, the optimization problem is listed.In the fourth section, the PPO2 algorithm is used to solve the optimization problem.The fifth section is the experimental part. By changing the relevant parameters, we can observe the communication times, model accuracy, and model The simulation results show the convergence of the PPO2 algorithm.The last section of this paper is the summary, which summarizes the main points and shortcomings of this work.

## 2. Related Work

The bottleneck limiting the performance of the AUV swarm mainly lies in the difficulties of underwater communications [6]. When AUV groups are used for federated learning tasks, as the scale of task models and AUV groups expand, the pressure on the self-organizing data interaction network increases, which requires designing algorithms to save communication resources as much as possible. However, reducing communication resources excessively may lead to compromised federated learning performance and confusion about the AUV formations, which creates a paradox. How to make a trade-off between resource consumption and model performance is worth studying at present.

AUV swarm has the advantages of rapid deployment, controllable action, flexible networks, and other advantages because of its formation, dynamic, mobility, and other characteristics. In recent years, the research into underwater exploration systems based on multiple AUVs or AUV groups has become popular, especially in navigation or path planning [7], collaborative data collection [8], target hunting [9], and so on. For example, in the offshore oil and gas industry, AUVs equipped with underwater environment monitoring sensors cooperate to effectively detect the boundary of oil and gas producing areas. Cui et al. [10] proposed an adaptive path planning algorithm based on the random tree star algorithm to estimate the scalar field underwater for the AUV group. Noguchi et al. [11] put forward a dynamic task allocation and path planning algorithm for the AUV group, which gives full play to the advantages of the AUV group. Huang et al. [12] used the AUV group equipped with scanning imaging sonar, and proposed using the binary Bayesian filter to process the input signal. Then, a path-planning strategy based on the artificial potential field was proposed for real-time target tracking. Cao et al. [13] used the artificial potential field theory to model the underwater environment information and introduce the potential field function into the dispersion, similarity, and difference of the potential field function. On this basis, an intelligent path planning model that can achieve accurate AUV swarm collaborative target search is proposed.

In the federated learning aggregation algorithm, each participant needs to completely send local parameters to the server for model aggregation. These model parameters tend to have a lot of data, which brings huge communication overhead. In [14], the author protects the privacy of the federated learning architecture through covert communication technology, minimizes the federated learning delay by optimizing the interference power, signal power, and training accuracy, and uses an alternating descent algorithm to solve the optimization problem. In [15], the author uses federated learning to build a digital twin model based on the digital twin edge network architecture, uses an asynchronous model update scheme, and performs device selection locally, minimizing channel allocation, and CPU frequency as variables. The federated learning delay uses the artificial intelligence method DNN (deep neural network) model to dynamically solve the optimization problem. In [16], the author selects some clients to send their local model parameters to the server, avoids the training loss, and convergence time being affected by a reasonable local selection strategy, and establishes an optimization problem to optimize the convergence time and training loss, using ANN (artificial neural network) to solve optimization problems. In [17], the author defines a new performance evaluation criterion, i.e., learning efficiency, which is discussed in scenarios where the device is equipped with CPU and GPU, respectively, e.g., channel resource allocation and bandwidth allocation as optimization variables were used to maximize learning efficiency via the Lagrange multiplier method, KKT (Karush–Kuhn–Tucker) conditions. A two-dimensional search algorithm was used to solve the optimization problem and detect and skip some redundant communication rounds locally with adaptive communication rules; a comparable convergence speed was used to the original SGD.

In [18], the authors proposed a communication-saving method for distributed machine learning, which is a gradient-based selection method that reduces redundant communication rounds. In [19], the author introduced the above method [18] into federated learning, based on the traditional SGD (stochastic gradient descent) method. In [20], the author delves into the basis of the above work, and used the gradient sparse method to select redundant nodes locally to reduce the communication amount. The work of this paper continues on this basis. So far, there is no work to introduce federal learning into AUV swarm.

## 3. System Model

This paper considers the following scenario: a fixed-form swarm consisting of a leader AUV *L* and a set M of the *M* follower AUV sailing at the same depths with constant speed, collecting seabed data, cooperating, and using federated learning to perform various machine learning tasks, such as target recognition, path planning, etc. Each AUV maintains the same distance from other AUVs and navigators, collecting data with their own independent sensors, and training local models. The local model parameters are then sent to the navigator using the uplink. The leader AUV uses the information uploaded by the follower to aggregate into global model parameters and sends it to the follower using the downlink. The follower proceeds with a new round of local training using the received global model parameters. In order to keep AUVs in formation, AUV *L* and AUV *m* also need to transfer position information and speed information to each other. In each round, AUV *m* uploads its current speed and position data to AUV *L*. AUV *L* calculates and broadcasts the subsequent speed and direction from the received information.

### 3.1. Federated Learning Model

Assuming that the input data collected by the follower AUV m∈M is $X_{m}$, the output data are $Y_{m}$, and the local model parameter is $w_{m}$, then the data set of the AUV *m* can be expressed as $D_{m}=\left\{\left(x_{m,1},y_{m,1}\right),\left(x_{m,2},y_{m,2}\right),\ldots ,\left(x_{m,N_{m}},y_{m,N_{m}}\right)\right\}$, where $N_{m}$ is the number of the data that AUV *m* owns, the loss function on its data set $D_{m}$ can be expressed as average of sample loss functions:(1)$$F_{m}\left(w_{m}\right)=\frac{1}{N_{m}}\sum_{i=1}^{N_{m}}f\left(w_{m};x_{m,i},y_{m,i}\right),\;\forall m\in M$$

Assuming that the global model parameter is w, for an AUV m∈M containing $N_{m}$ data, the corresponding local loss function is weighted, averaged, and defined as the global loss function:(2)$$F\left(w\right)≜\sum_{m=1}^{M}\frac{N_{m}F_{m}\left(w\right)}{N}=\frac{1}{N}\sum_{m=1}^{M}\sum_{i=1}^{N_{m}}f\left(w_{m};x_{m,i},y_{m,i}\right),$$

The purpose of federated learning is to find a parametric model that minimizes the global loss function, i.e., the optimal parametric model can be expressed as
(3)$$w^{*}=\arg\;\min\;F\left(w\right),$$
where $w=w_{1}=\cdots =w_{M}$,

### 3.2. Communication Model

In the underwater environment, electromagnetic waves and optical signals attenuate quickly and travel over short distances. Therefore, the most commonly used underwater communication is underwater acoustic communication. The attenuation of the water signal with frequency *f* at distance *D* can be given by
(4)$$A\left(D,f\right)=l^{k}a{\left(f\right)}^{D}$$
where *k* represents the spreading factor, and a(f) is the absorption coefficient, which can be expressed by the following empirical formula [21]:(5)$$10\log a\left(f\right)=\frac{0.11f^{2}}{1+f^{2}}+\frac{44f^{2}}{4100+f^{2}}+2.75\cdot 10^{-4}f^{2}+0.003.$$

In the marine environment, noise is mainly divided into turbulence noise, shipping noise, wave noise, and thermal noise, respectively. According to [22], the power spectral density (p.s.d) of four main types of noise in dB re μPa per Hz on the communication frequency *f* can be given by:(6)$$10\log N_{\vartheta }\left(f\right)=17-30\log f$$
(7)$$10\log N_{s}\left(f\right)=40+20\left(s-\frac{1}{2}\right)+26\log f-60\log\left(f+0.03\right)$$
(8)$$10\log N_{w}\left(f\right)=50+7.5w^{\frac{1}{2}}+20\log f-40\log\left(f+0.4\right)$$
(9)$$10\log N_{th}\left(f\right)=-15+20\log f$$

Moreover, s∈[0,1] is the shipping activity factor, while *w* represents the wind velocity (m/s). Hence, the combined noise N(f) can be represented as
(10)$$N\left(f\right)=N_{\vartheta }\left(f\right)+N_{s}\left(f\right)+N_{w}\left(f\right)+N_{th}\left(f\right)$$

Therefore, the normalized SNR of a signal with unit transmitted power and bandwidth can be represented as
(11)$$\gamma \left(D,f\right)=\frac{1}{A(D,f)N(f)}.$$

The uplink data rate between AUV *m* and leader AUV *L* is given by
(12)$$R_{m}^{U}=B_{m}^{U}\log_{2}\left(1+\frac{p_{m}\gamma \left(D,f\right)}{B_{m}^{U}}\right),$$
where $B_{m}^{U}$ is the allocated uplink bandwidth of the AUV *m*, while $p_{m}\in \left(0,p_{max}\right)$ is the transmitting power of the AUV *m*.

In the iteration, the leader AUV broadcasts the information to the followers and, hence, the downlink data rate between AUV *m* and leader AUV *L* can be represented as
(13)$$R_{m}^{D}=B^{D}\log_{2}\left(1+\frac{p_{L}\gamma \left(D,f\right)}{B^{D}}\right)$$
where $B^{D}$ is the downlink bandwidth, $p_{L}\in \left(0,p_{max}\right)$ denotes the transmitting power of leader AUV.

### 3.3. Control Model

Due to the high cost of underwater communication, most gradient interactions in the distributed SGD are redundant. Therefore, this paper introduces the concept of lazy nodes, allowing each follower drone to perform self-detection locally so that some nodes skip some rounds of communication. The definition of a lazy node in this paper refers to the work of Chen et al. [19].
(14)$$\frac{{∥\nabla_{M_{N}}^{t-1}∥}^{2}}{M_{N}}⩽\frac{{∥\nabla_{M}^{t-1}∥}^{2}}{M}$$

Among them, these lazy nodes constitute a set $M_{N}$ of size $M_{N}$. In this paper, the gradient descent algorithm is used to optimize the global model, where γ represents the learning rate
(15)$$w\left(t\right)=w\left(t-1\right)-\gamma \nabla_{M}^{t-1}$$
(16)$${∥\nabla_{M}^{t-1}∥}^{2}⩽\frac{M_{N}}{\gamma^{2}M}{∥w\left(t\right)-w\left(t-1\right)∥}^{2},$$

Since the global model tends to converge, the following approximation is used
(17)w(t)−w(t−1)≈w(t−1)−w(t−2)

According to the mean inequality, we have
(18)$${∥\nabla_{M_{N}^{t-1}}∥}^{2}={∥\sum_{m\in M_{N\;}}\frac{N_{m}\nabla_{m}^{t-1}}{N}∥}^{2}⩽\frac{M_{N\;}}{N^{2}}\sum_{m\in M_{N\;}}{∥N_{m}\nabla_{m}^{t-1}∥}^{2}$$

Let $M_{N}=\beta M$, Equation (Equation 14) can be deduced as Equation (Equation 18)
(19)$${∥N_{m}\nabla_{m}^{t-1}∥}^{2}⩽\frac{N^{2}}{\gamma^{2}M^{2}\beta }{∥w\left(t-1\right)-w\left(t-2\right)∥}^{2}.$$

At one round *t*, AUV *m* checks locally whether Equation (Equation 18) is satisfied. If satisfied, skip this round of uploading, otherwise participate in this round of uploading. Considering the extreme case, if all nodes in a certain round *t* satisfy Equation (Equation 18), then AUV *L* cannot receive the model information uploaded by the follower AUV. At this time, AUV *L* randomly selects some follower drones to participate in the upload after a certain time interval. In this paper, we randomly select 1 AUV *m* to participate in uploading in extreme cases.

In order to prevent AUVs from not communicating for a long time due to the setting of lazy nodes at certain times, we require that AUVs communicate at least once within τ round. Each AUV locally records whether it uploads itself in each round of uploading
(20)$$\Lambda_{m}\left(\tau \right)=\left\{\lambda_{m}\left(t-\tau \right),\lambda_{m}\left(t+1-\tau \right),\ldots ,\lambda_{m}\left(t-1\right)\right\}$$
where $\lambda_{m}\left(t\right)\in \left\{0,1\right\}$ indicates whether AUV *m* participates in the *t* round of gradient upload after controlling the model, $\lambda_{m}\left(t\right)$ = 1 indicates that it participates in this round of the gradient upload, otherwise not.

At one round *t*, AUV *m* checks locally whether Equation (Equation 20) is all of 0. If so, AUV *m* must be uploaded in this round.

### 3.4. Latency Model

#### 3.4.1. Local Parameter Calculating Latency

Let $f_{m}\in \left(f_{\min},f_{\max}\right)$ and $f_{L}\in \left(f_{\min},f_{\max}\right)$ denote the CPU frequency of the AUV *m* and leader AUV *L*, respectively, where $f_{\min}$ and $f_{\max}$ represent the minimum and the maximum CPU frequency, respectively.

The local computation latency on each AUV *m* at the *t*-th slot can be calculated as
(21)$$T_{m}^{\mathrm{LC}}\left(t\right)=\frac{N_{m}c_{m}}{f_{m}\left(t\right)},$$
where $c_{m}$ is the required CPU cycles for training one sample data by backpropagation algorithm [17] at AUV *m*.

#### 3.4.2. Uploading Latency

Assume that the local parameter has a counterpart gradient consists of α elements, and each of them has an average quantitative bit number denoted by ζ. The size of the current position information and speed information is ψ. Hence, the total data size of each local parameter is $\left|w_{m}\right|=\alpha \zeta$ [17], uploading latency is given by
(22)$$T_{m}^{\mathrm{LU}}\left(t\right)=\frac{\left|w_{m}\right|\lambda_{m}\left(t\right)}{R_{m}^{U}}+\frac{\psi \lambda_{m}\left(t\right)}{R_{m}^{U}}$$

#### 3.4.3. Global Parameter Aggregating Latency

At the leader AUV *L*, the global parameter aggregating latency is calculated by
(23)$$T_{L}^{\mathrm{GA}}\left(t\right)=\frac{c_{0}\sum_{m=1}^{M}\left|w_{m}\right|}{f_{L}\left(t\right)},$$
where $c_{0}$ is the computational complexity [15] to aggregate the newly updated parameters from follower AUVs, while $\left|w_{m}\right|$ is the data size of the local parameter $w_{m}$.

#### 3.4.4. Global Parameter Updating Latency

The local parameter updating latency is given by
(24)$$T_{L}^{\mathrm{GU}}\left(t\right)=\frac{c_{L}^{\prime }}{f_{L}\left(t\right)},$$
where $c_{L}^{\prime }$ is the computational complexity for performing global parameter updating [17].

#### 3.4.5. Downloading Latency

The downloading latency can be calculated as
(25)$$T_{L}^{\mathrm{GD}}\left(t\right)=\frac{\left|w\right|}{R_{m}^{D}}+\frac{\psi }{R_{m}^{D}},$$
where $\left|w\right|$ is the data size of the global parameter, which is basically similar to the local parameter $\left|w_{m}\right|$.

#### 3.4.6. Total Latency

Regardless of extreme cases, there are always devices uploaded in a round, a complete federated learning delay is
(26)$$T\left(t\right)=\max_{m\in M}\left\{T_{m}^{\mathrm{LC}}+T_{m}^{\mathrm{LU}}\right\}+\max_{m\in M}\left\{T_{L}^{\mathrm{GA}}+T_{L}^{\mathrm{GU}}+T_{L}^{\mathrm{GD}}\right\}.$$

Consider the extreme case where there are no devices uploaded within the round. According to the settings of the control model, when the AUV *L* does not receive the gradient upload after the time interval $T^{\prime }$, a device is randomly selected for uploading. Assuming that the selected device is $m_{0}$, then the federated learning delay for this round is
(27)$$T\left(t\right)=T_{m_{0}}^{\mathrm{LC}}+T_{m_{0}}^{\mathrm{LU}}+\max_{m\in M}\{T_{L}^{\mathrm{GA}}+T_{L}^{\mathrm{GU}}+T_{L}^{\mathrm{GD}}\}+T^{\prime }.$$

### 3.5. Energy Consumption Model

#### 3.5.1. Energy Consumption of Follower AUV

The computational energy consumption of the follower AUV is mainly composed of a local parameter calculating and local parameter updating, which can be given by
(28)$$E_{m}^{\mathrm{Cp}}\left(t\right)=kf_{m}^{\sigma }\left(T_{m}^{\mathrm{LC}}\left(t\right)\right),$$

The communication energy consumption is mainly the local parameter uploading, and it can be calculated by
(29)$$E_{m}^{C}\left(t\right)=p_{m}T_{m}^{\mathrm{LU}}\left(t\right),$$

#### 3.5.2. Energy Consumption on Leader AUV

The computational energy consumption of the leader AUV is mainly caused by a global parameter aggregating, which is given as
(30)$$E_{L}^{\mathrm{Cp}}\left(t\right)=kf_{L}^{\sigma }\left(T_{L}^{\mathrm{GA}}\left(t\right)+T_{L}^{\mathrm{GU}}\left(t\right)\right),$$

Similarly, the communication energy consumption of leader AUV is mainly due to the global parameter downloading, which can be calculated by
(31)$$E_{m}^{C}\left(t\right)=p_{L}T_{L}^{\mathrm{GD}}\left(t\right),$$

#### 3.5.3. Total Energy Consumption


(32)
$$E\left(t\right)=\Phi \left(E_{m}^{\mathrm{Cp}}\left(t\right)+E_{L}^{\mathrm{Cp}}\left(t\right)\right)+\chi \left(E_{m}^{C}\left(t\right)+E_{m}^{C}\left(t\right)\right)$$


### 3.6. Problem Formulation

Our goal is to minimize the total cost by optimizing the variables $p_{m},f_{m},p_{L},f_{L},\beta$, which are defined as follows.
(33)$$\begin{array}{c} \begin{array}{cc} P1: & \min_{p_{m},f_{m},p_{L},f_{L},\beta }\mathrm{Cost}=\sum_{t=1}^{N_{t}}\left(T\left(t\right)+E\left(t\right)\right)\; \end{array} \end{array}$$
(34)$$0⩽p_{m}⩽p_{\max},m\in M,$$
(35)$$f_{\min}⩽f_{m}⩽f_{\max},m\in M,$$
(36)$$0⩽p_{L}⩽p_{\max},$$
(37)$$f_{\min}⩽f_{L}⩽f_{\max},$$
(38)0⩽β⩽1,
(39)$$E_{m}^{\mathrm{Cp}\;}+E_{m}^{C\;}⩽E_{m}^{\mathrm{thd}\;},m\in M,$$
(40)$$E_{L}^{\mathrm{Cp}\;}+E_{L}^{C\;}⩽E_{L}^{\mathrm{thd}\;},$$

Among them, $N_{t}$ is the total number of iterations. Equation (Equation 33) is the optimization objective, and the total time of each round is minimized by the optimizing power $p_{m},p_{L}$ and CPU frequency $f_{m},f_{L}$, Equations (Equation 34) and (Equation 35) are the power constraints, and CPU frequency constraints of the AUV *m*, respectively, Equations (Equation 36) and (Equation 37) are the power constraints and CPU frequency constraints of AUV *L*, respectively, Equation (Equation 38) is the lazy node scale factor, and Equations (Equation 39) and (Equation 40) are the energy constraints of AUV *m* and AUV *L*, respectively.

## 4. Algorithm Design

In this section, we consider PPO to deal with P1, since it is non-convex and has high dimensionality.

### 4.1. Modeling of Deep Reinforcement Learning Environment

An MDP consists of the state space, action space, state transition matrix function, reward function, and discount factor. Thus the optimization problem P1 can be transformed into the following:

**State space**: For each time slot *t*, we use $p_{m}\left(t-1\right),f_{m}\left(t-1\right),p_{L}\left(t-1\right),f_{L}\left(t-1\right),\beta \left(t-1\right)$ at the time slot (t−1) to describe the state space.

According to the above, the network state of the agent at time slot *t* in this paper can be expressed by
(41)$$\begin{array}{c} \begin{array}{c} s\left(t\right)=\left(p_{m}\left(t-1\right),f_{m}\left(t-1\right),p_{L}\left(t-1\right),f_{L}\left(t-1\right),\beta \left(t-1\right)\right) \end{array} \end{array}$$
the network state of the agent is given by $S_{t}=\left\{s(t)\right\}$

**action space**: at each the time slot *t*, a(t) is composed of the following parts—formally, the action space at time slot *t* is denoted by: $p_{m}\left(t\right),f_{m}\left(t\right),p_{L}\left(t\right),f_{L}\left(t\right),\beta \left(t\right)$
(42)$$\begin{array}{c} \begin{array}{c} a\left(t\right)=\left(p_{m}\left(t\right),f_{m}\left(t\right),p_{L}\left(t\right),f_{L}\left(t\right),\beta \left(t\right)\right) \end{array} \end{array}$$
and the action of the agent is given by $a_{t}=\left\{a(t)\right\}$.

**State transition function**: transition probability of the agent at time slot *t* can be denoted as $P_{T}\left(s_{t+1}|s_{t},a_{t}\right)$.

**Policy**: Let π denote the policy function, which is based on the observed state to make decisions and control the action of the agent π(a∣s)=P(a∣s).

**Reward function**: the reword function in this paper is designed for optimizing $p_{m},f_{m},p_{L},f_{L},\beta$; the presented reword function of the agent at one time slot *t* is expressed as
(43)$$\begin{array}{c} r_{t}\left(s_{t},a_{t}\right)=-\left(T\left(t\right)+E\left(t\right)\right) \end{array}$$

Maximizing the cumulative reward obtained by the sequence $T^{\prime }$ involves the sum of the rewards obtained at each stage, called $R_{n}\left(T^{\prime }\right)$. Therefore, the expected reward can be obtained as follows with policy π:(44)$$\begin{array}{c} \begin{array}{c} R_{n}\left(T^{\prime }\right)=\sum_{\tau =0}^{\infty }\xi_{\tau }r_{n}\left(s_{n+\tau },a_{n+\tau }\right) \end{array} \end{array}$$
where $\xi_{\tau }$ denotes the discounter factor.

### 4.2. Proximal Policy Optimization Algorithm

The proximal policy optimization (PPO) is the off-policy model-free reinforcement learning algorithm. OpenAI uses it as the current baseline algorithm, which uses a new class of objective functions and updates parameters in small batches with multiple training steps. To better describe reward and prevent overfitting, we used advantage ${\hat{A}}_{n}$ instead $R_{n}\left(T^{\prime }\right)$ to evaluate actions
(45)$$\begin{array}{c} \begin{array}{c} {\hat{A}}_{n}=R_{n}\left(T^{\prime }\right)-V_{\phi }\left(s_{n}\right) \end{array} \end{array}$$
where $V_{\phi }\left(s_{n}\right)$ can be calculated by a value network.

Moreover, our expectation is to update the actor’s policy π to maximize the expected reward. So, we need to use the gradient boosting method to update the network parameters θ. The gradient solution process is as follows:(46)$$\begin{array}{c} \begin{array}{c} G=E\left[\nabla_{\theta }\log\pi_{\theta }\left(a_{t}|s_{t}\right){\hat{A}}_{t}\right] \end{array} \end{array}$$

PPO2 was inspired by the same question as TRPO: using only the current data to improve the policy as much as possible, without causing a sudden decline in the performance of the policy. Differently from TRPO attempting to solve the problem with a complex second-order approach, PPO2 uses a first-order approach that uses a few other tricks to make the new policy approximate the old one. The essence of the PPO algorithm is to introduce a ratio coefficient to indirectly describe the difference between the new strategy and the old strategy, denoted by $r_{t}\left(\theta \right)=\frac{\pi_{\theta }\left(a_{t}|s_{t}\right)}{\pi_{\theta_{k}}\left(a_{t}|s_{t}\right)}$; the loss function is defined as follows:(47)$$\begin{array}{c} \begin{array}{cc} & \theta_{k+1}=\arg\max_{\theta }\frac{1}{\left|D_{k}\right|T}\sum_{\tau \in D_{k}}\sum_{t=0}^{T}\\ & \min\left(\frac{\pi_{\theta }\left(a_{t}|s_{t}\right)}{\pi_{\theta_{k}}\left(a_{t}|s_{t}\right)}A^{\pi_{\theta_{k}}}\left(s_{t},a_{t}\right),\;g\left(\epsilon ,A^{\pi_{\theta_{k}}}\left(s_{t},a_{t}\right)\right)\right) \end{array} \end{array}$$
where ϵ is a (small) hyperparameter that roughly limits the range of variation of $r_{t}\left(\theta \right)$. In PPO2, we use the following formula to update the value network strategy
(48)$$\begin{array}{c} \begin{array}{c} \phi_{k+1}=\arg\min_{\phi }\frac{1}{\left|D_{k}\right|T}\sum_{\tau \in D_{k}}\sum_{t=0}^{T}{\left(V_{\phi }\left(s_{t}\right)-{\hat{R}}_{t}\right)}^{2} \end{array} \end{array}$$

The algorithm used in this paper is summarized in Algorithm 1.
**Algorithm 1** Proximal policy optimization clip.Input: initial policy parameters $\theta_{0}$, initial value function parameters $\phi_{0}$**for** k=1,2,…**do**   Run policy $\pi_{\theta }$ for *K* timesteps, collecting $s_{n},a_{n},r_{n}$   Compute return ${\hat{R}}_{n}$ according to Equation (Equation 44)   Compute advantages ${\hat{A}}_{n}$ according to Equation (Equation 45)   Update the policy by maximizing the PPO clip objective according to Equation (Equation 47)   Fit value function according to Equation (Equation 48)**end for**

## 5. Simulation Results

This chapter uses the TensorFlow framework in Python to build a federated learning model, using a MNIST (Modified National Institute of Standards and Technology Database) consisting of 42,000 digital images, and retaining 10% of the data as a test set for the global model. At the same time, a three-layer MLP (multilayer perceptron) neural network was created as a model for the classification task. The number of neurons in each layer was 200, 200, and 10, respectively. We used this model to perform a classification task and classify the resulting accuracy and loss function. This section gives a maximum number of convergence rounds of 1000. Assuming *M* = 9, the follower AUVs are 20 m apart from each other and form a formation to form a formation and fly at a constant speed in a certain direction, and the leader AUV is 20 m in front of the formation and the center. The rest of the simulation parameters are shown in Table 1.

### 5.1. The Performance of Each Index in the Gradient Compression Test

In Figure 1, the comparison of communication times among the different schemes versus β, We can observe that as β increases, the follower AUVs that do not participate in communication within the round decrease more, and the corresponding total number of communications increased. From the analysis of Equation (Equation 19), the decrease of β leads to the increase of the right side of the inequality, resulting in more nodes satisfying Equation (Equation 19), the AUV skipping communication rounds increase, and the communication decreases. The experimental results are in good agreement with the theoretical analysis. It can be concluded from Figure 1 and Figure 2 that the federated learning after the control model still converges. However, as it decreases, the number of communications decreases. Although the communication resources are saved, the global model is affected due to the lack of gradients of some follower AUVs in certain rounds, resulting in a decrease in the performance of federated learning. Specifically, the performance is reduced in accuracy, and the experimental results are consistent with the theoretical analysis.

In Figure 3, we show the total time comparison of different schemes versus β. We can observe that as β increases, the total time reduces. This is because with β increasing, the communications gradually increase, and the model can converge faster with more communications, resulting in a shorter total time.

### 5.2. The Performance Analysis of the Scheme Proposed in This Paper

To show that the scheme proposed in this paper has the best effect on reducing the training cost, we compare it with other offloading schemes.
**Scheme 1**: the scheme proposed in this paper.**Scheme 2**: asynchronous federated learning with dynamically optimized β**Scheme 3**: asynchronous federated learning with fixed β.**Scheme 4**: asynchronous federated learning with LAG.**Scheme 5**: traditional asynchronous federated learning.

The two sub-graphs in Figure 4 describe the communication times and corresponding accuracy rates of different experimental schemes with different follower AUV numbers. Accuracy will increase. At the same time, a larger communication number corresponds to a higher accuracy rate. When the communication number time is reduced by 710, which is only 21% of the traditional asynchronous federated learning, the accuracy rate is only reduced by 0.03%. This shows that we greatly reduced the communication number, but the accuracy basically did not decrease, indicating that our proposed scheme has good results.

In Figure 5, we show the accuracies of different control models versus the number of follower AUVs. Compared with the work of [19], the control model proposed in this paper increases the accuracy. This shows that the improved control model in this paper is effective.

In Figure 6, we show the cost comparison of different schemes versus the number of follower AUVs. With the increase in the follower AUV, the cost gradually increases. As can be seen from the figure, the scheme proposed in this paper has a smaller cost and can save more resources.

### 5.3. The Performance Analysis of the PPO2 Algorithm

In Figure 7, we compare the performance of state-of-the-art algorithms in coping with problem P1, including GA, PSO, AC, DDPG, and our employed PPO2 algorithm. We can observe that PPO2 has a smaller cost under the same number of iterations, which shows that this algorithm is better than other algorithms.

## 6. Conclusions

In order to reduce the constraints of underwater scarce communication resources on AUV swarm machine learning, we designed an asynchronous federated learning method. By constructing the optimization problem of minimizing the weighted sum of delay and energy consumption, the AUV CPU frequency and signal transmission power were jointly optimized. In order to solve this complex optimization problem of high-dimensional non-convex time series accumulation, we transformed the problem into an MDP and used the PPO2 algorithm to solve this problem. Finally, we carried out some experiments to verify the effectiveness of the proposed scheme.

## Figures and Tables

**Figure 1 sensors-22-08727-f001:**
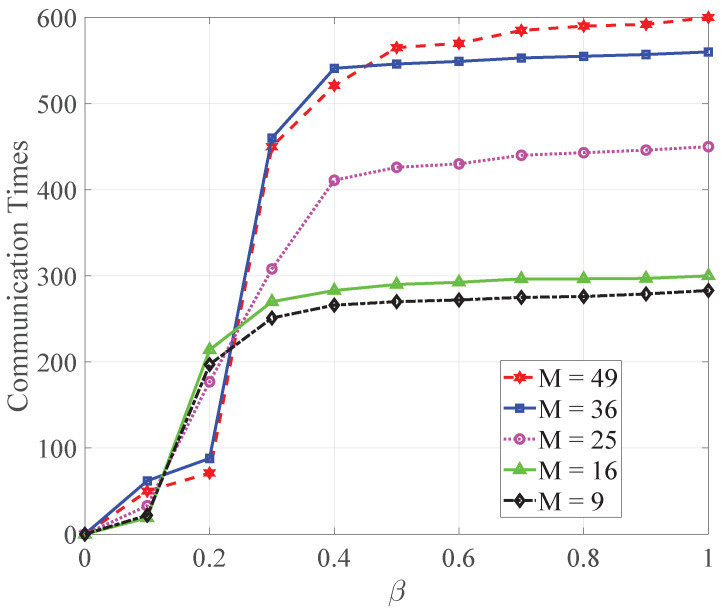
Comparison of the communication times of different schemes versus β.

**Figure 2 sensors-22-08727-f002:**
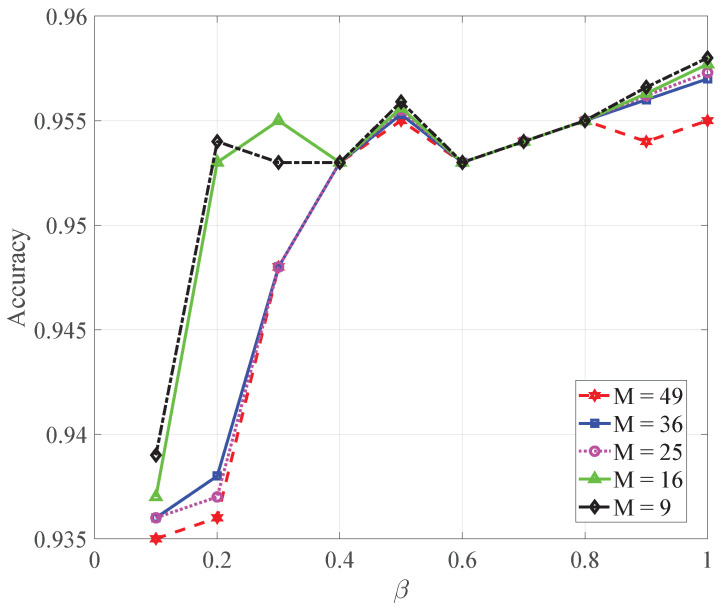
Comparison of the accuracy of different schemes versus β.

**Figure 3 sensors-22-08727-f003:**
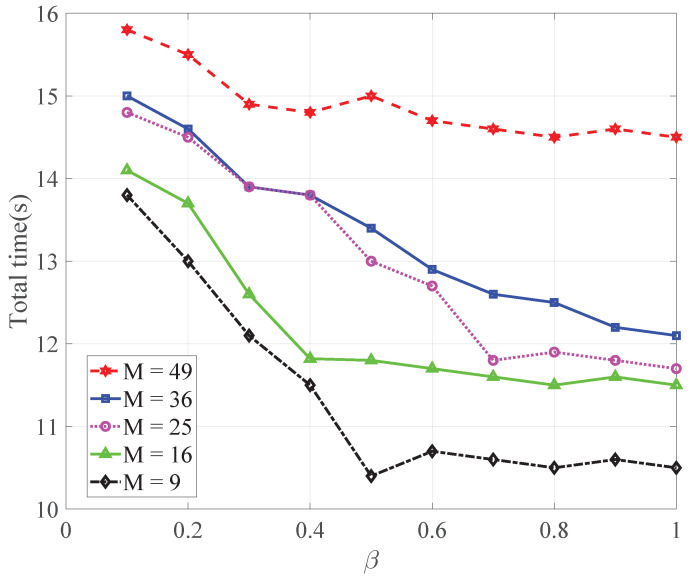
Comparisons of the total times of different schemes versus β.

**Figure 4 sensors-22-08727-f004:**
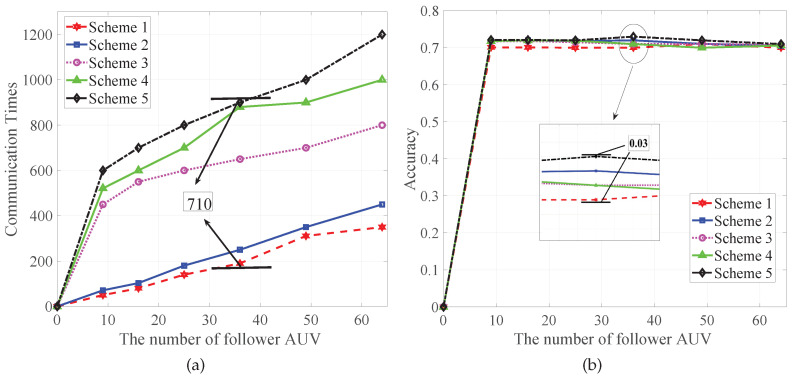
The relationship between communication times and accuracy. (**a**) Comparison of the communication times of different schemes versus the number of follower AUVs. (**b**) Comparison of the accuracy of different schemes versus the number of follower AUVs.

**Figure 5 sensors-22-08727-f005:**
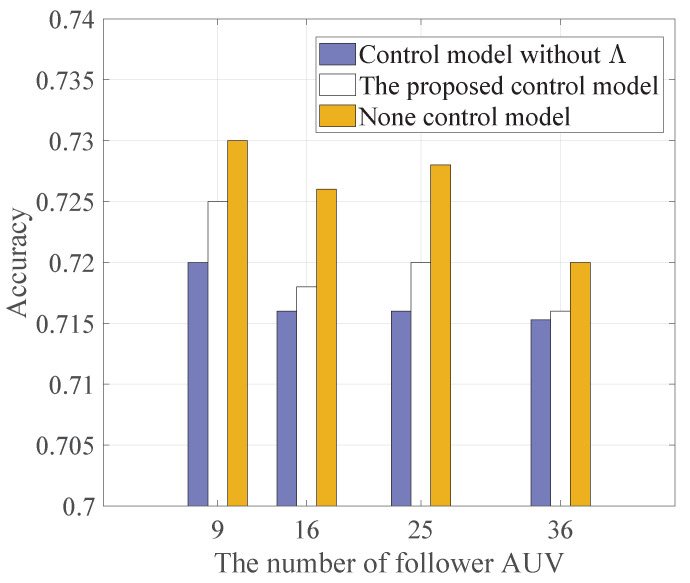
Comparison of the accuracies of different control models versus the number of follower AUVs.

**Figure 6 sensors-22-08727-f006:**
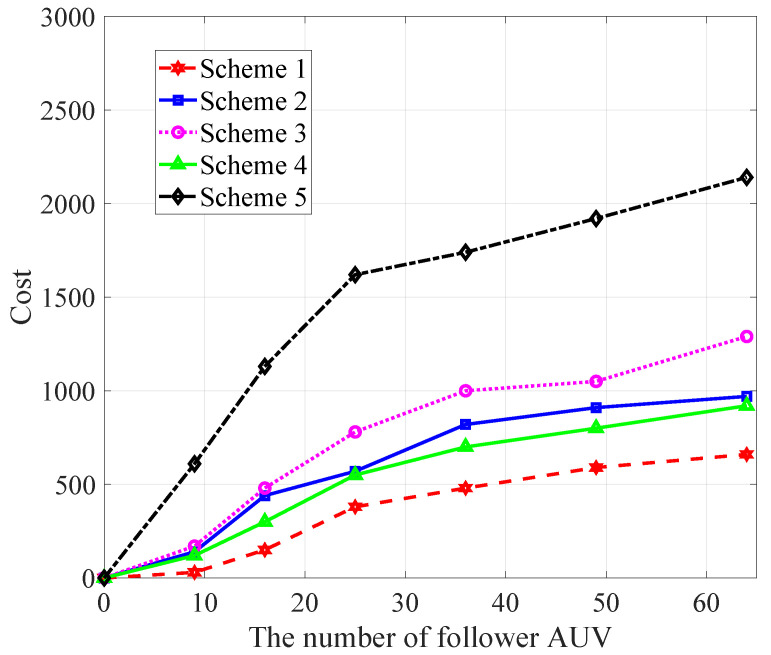
Comparison of the costs of different schemes versus β.

**Figure 7 sensors-22-08727-f007:**
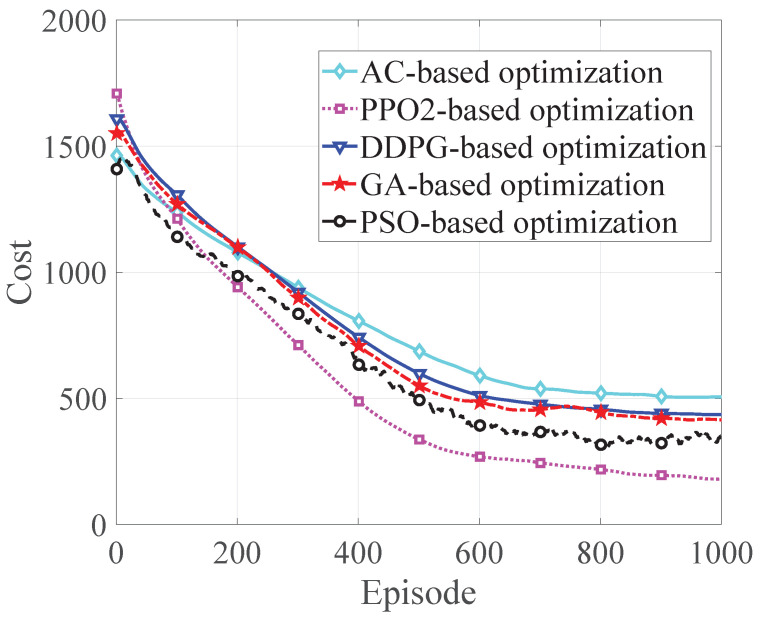
Comparison of the profits of different algorithms.

**Table 1 sensors-22-08727-t001:** Values of main parameters.

Parameter	Value	Parameter	Value
*k*	$1.25\times 10^{-26}$	γ	0.01
δ	3	M	20
$B_{m}^{U}$	10 kHz	*f*	30 kHz
$B^{D}$	10 kHz	s	0.5
$N_{m}$	4224	τ	5
$c_{m}$	10,000	$f_{\max}$	0.4 GHz
$c_{0}$	50	$f_{\min}$	0.2 GHz
$c_{L}^{\prime }$	50	$E_{m}^{\mathrm{thd}}$	0.07 W
$|w_{m}|$	1594*64 bit	$E_{L}^{\mathrm{thd}}$	0.8 W
|w|	1594*64 bit	Φ	1
$p_{\max}$	0.2 W	χ	1

## Data Availability

Not applicable.

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
