# Peer review of "Efficient Asynchronous Federated Learning for AUV Swarm"

_sensors, 2022, doi:10.3390/s22228727_

Round 1
Reviewer 1 Report
See attachment.

Reviewer 2 Report
The paper is generally well-written. On the other hand, the paper should be revised by considering the following issues. MAJOR ISSUES: 1) The related work of the paper should be improved by citing more papers. There are only 18 references which is very low for a journal paper. Some other papers tackles similar trajectory planning problems for UAVs and robot swarms by considering time and energy efficiency. For this purpose, I strongly recommend the authors should include the following two papers in their related work in order to clarify not only the main contribution but also motivation of proposed approach in this paper in the related literature.
- O. M. Gul, A. M. Erkmen, and B. Kantarci, "UAV-Driven Sustainable and Quality-Aware Data Collection in Robotic Wireless Sensor Networks", IEEE Internet of Things Journal (early access). - O. M. Gul, A. M. Erkmen, "Energy-Efficient Cluster-Based Data Collection by a UAV with a Limited-Capacity Battery in Robotic Wireless Sensor Networks", Sensors, vol.20, no.20,Sep. 2020. 2) As suggestion, the authors may also consider the work with orienteering problem with single collector and multiple-collectors (vehicle routing problem) for more idea and similar problems. They may check the following survey for orienteering problem. - Vansteenwegen, P. , Souffriau, W. , and Van Oudheusden, D., ”The orienteering problem: A survey”, European Journal of Operational Research, 209 (1), pp. 1-10, 2011.
3) The outline of the paper should be given at the end of the introduction section.
MINOR ISSUES
+Size of Figures should be enlarged. +Typos and grammatical errors should be fixed.
Round 2
Reviewer 2 Report
The paper is generally well-written.
On the other hand, the paper should be revised by considering my comments (in fact, I shared them in the previous round):
MAJOR ISSUES:
1) The related work of the paper should be improved by citing more papers. There are only 18 references which is very low for a journal paper. Some other papers tackles similar trajectory planning problems for UAVs and robot swarms by considering time and energy efficiency. For this purpose, I strongly recommend the authors should include the following two papers in their related work in order to clarify not only the main contribution but also motivation of proposed approach in this paper in the related literature.
- O. M. Gul, A. M. Erkmen, and B. Kantarci, "UAV-Driven Sustainable and Quality-Aware Data Collection in Robotic Wireless Sensor Networks", IEEE Internet of Things Journal (early access).
- O. M. Gul, A. M. Erkmen, "Energy-Efficient Cluster-Based Data Collection by a UAV with a Limited-Capacity Battery in Robotic Wireless Sensor Networks", Sensors, vol.20, no.20,Sep. 2020.
2) As suggestion, the authors may also consider the work with orienteering problem with single collector and multiple-collectors (vehicle routing problem) for more idea and similar problems. They may check the following survey for orienteering problem. - Vansteenwegen, P. , Souffriau, W. , and Van Oudheusden, D., ”The orienteering problem: A survey”, European Journal of Operational Research, 209 (1), pp. 1-10, 2011.
3) The outline of the paper should be given at the end of the introduction section.
MINOR ISSUES
+Size of Figures should be enlarged.
+Typos and grammatical errors should be fixed.
